# Effect of Deformation Parameters of an Initial Aged GH4169 Superalloy on Its Microstructural Evolution during a New Two-Stage Annealing

**DOI:** 10.3390/ma15165508

**Published:** 2022-08-11

**Authors:** Mingsong Chen, Quan Chen, Yumin Lou, Yongcheng Lin, Hongbin Li, Guanqiang Wang, Hongwei Cai

**Affiliations:** 1Light Alloy Research Institute, Central South University, Changsha 410083, China; 2State Key Laboratory of High Performance Complex Manufacturing, Changsha 410083, China; 3School of Mechanical and Electrical Engineering, Central South University, Changsha 410083, China; 4Zheneng Technology Research Institute Co., Ltd., Hangzhou 310026, China; 5College of Metallurgies and Energy, North China Science and Technologies University, Tangshan 063009, China

**Keywords:** GH4169 superalloy, deformation parameters, new two-stage annealing, δ phase, microstructural evolution

## Abstract

This study aims to explore the effect of deformation parameters on microstructure evolution during the new two-stage annealing method composed of an aging treatment (AT) and a cooling recrystallization annealing treatment (CRT). Firstly, the hot compressive tests with diverse deformation parameters were finished for an initial aged deformed GH4169 superalloy. Then, the same two-stage annealing method was designed and carried out for the deformed samples. The results show that the deformation parameters mainly affect the grain microstructure during CRT by influencing the content, distribution and morphology of the δ phase after deformation. The reason for this is that there is an equilibrium of the content of the δ phase and Nb atom. When the deformation temperature is high, the complete dissolution behavior of the δ phase nuclei promotes the dispersion distribution of the δ phase with rodlike and needle-like shapes during AT. Thus, the fine and heterogeneous microstructure is obtained after annealing because the recrystallization nucleation is enhanced in those dispersed δ phases during CRT. However, when the retained content of δ phase nuclei is high after deformation, the clusters of intragranular δ phases will form during AT, resulting in the pinning of the motion for dislocation. The elimination of the mixed grain microstructure is slowed down due to the low static recrystallization (SRX) nucleation rate within the deformed grain.

## 1. Introduction

GH4169 superalloy is a crucial material for modern energy and aerospace industries owing to its fine corrosion resistance and prominent mechanical properties under high temperatures [1,2,3,4]. Thus, the key components in an aero-engine are usually made of GH4169 superalloy, such as turbine disks [5,6]. In general, hot forging is used in the formation of these important components [7]. Hence, the hot forming behaviors of GH4169 superalloy have been widely investigated. Lin et al. [8,9] researched their hot deformation behavior and established the constitutive modeling. Zhang et al. [10] analyzed the influence of deformation parameters on the microstructures, and modeled the correlation between these parameters and the grain size. Wen et al. [11] built the processing maps to obtain optimal deformation parameter domains. Additionally, Chen et al. [12] found that relatively large strains are essential for the full dynamic recrystallization (DRX) of GH4169 superalloy. Therefore, it is extremely difficult to obtain complete DRX during die forging, thereby causing the occurrence of a deformed microstructure comprising fine recrystallized grains and coarse deformed grains. Generally, the properties of components mainly depend on their microstructures [13,14,15], and a uniform and fine microstructure can effectively improve the properties [16,17]. Consequently, the mixed grain microstructure should be eliminated.

Owing to the existence of meta-dynamic recrystallization (MDRX) and SRX behaviors during annealing, the annealing treatment could serve as an efficient method to eliminate the mixed grain microstructure [18,19,20]. As a high-temperature stable phase, the δ phase (Ni_3_ Nb) is often used to pin grain boundary movement and act as nucleation sites for recrystallization at high temperatures [21,22,23,24]. In the authors’ previous work [25], it could be observed that the two-stage annealing treatment, which consists of an AT for precipitating some δ phases and a subsequent CRT to refine deformed mixed grains, is a valid way to eliminate the uneven deformed microstructure of an initial aged deformed GH4169 superalloy, and the mixed grains can be homogeneously refined to an average grain size of 5.82 μm under the routine of “900 °C × 12 h + 1010 °C − 2 °C/min − 970 °C”. However, the impact of deformation parameters on microstructure evolution during the new two-stage annealing process remains unclear. Generally, the diverse hot deformation parameters will cause the different initial deformed microstructures [26,27]. In addition, the initially deformed microstructures have an important influence on the microstructure evolution during annealing [28]. Hence, in-depth studies are essential to illustrate the influence of deformation parameters of an initial aged GH4169 superalloy on its microstructure evolution during the new two-stage annealing process.

In this study, for an initial aged GH4169 superalloy, a set of hot deformation experiments with varying deformation parameters were finished, and the two-stage annealing including AT and CRT was subsequently performed. Afterward, the role of different deformation parameters on the microstructure evolution during deformation was analyzed. Finally, the effects of deformation parameters on the evolution of the δ phase and grain microstructure during the two-stage annealing including AT and CRT were investigated.

## 2. Materials and Experiments

A commercial GH4169 superalloy was utilized, and its detailed chemical compositions (wt.%) are shown in Table 1. The size of experimental cylindrical specimens was ϕ10 mm × 15 mm. To dissolve all precipitates into the matrix and acquire homogenous initial microstructures, these samples were first solution-treated (T = 1040 °C, t = 45 mim). From Figure 1, it can be seen that almost no phase exists in the microstructure after the solution treatment, and only a little carbide exists in the matrix. Then, an aging treatment (T = 900 °C, t = 24 h) was performed to obtain sufficient δ phases. As displayed in Figure 2, there is an amount of long needle-like δ phases across grains, and the weight fraction of the δ phase reaches 23.09%. Afterward, the hot compressive experiments with diverse deformation parameters (as shown in Table 2) were carried out on the Gleeble-3500 machine (Poestenkill, NY, USA). The major cause for the choice of hot deformation parameters is that the typical uneven deformed microstructure can be obtained [29]. Eventually, for these diverse deformed samples, the two-stage annealing treatment composed of an AT and a CRT was implemented. Furthermore, the election of two-stage annealing parameters is based on the optimal annealing scheme in the authors’ previous investigation [25]. The detailed experimental process is illustrated in Figure 3.

The microstructure was characterized by optical microscope (OM), electron back-scattered diffraction (EBSD) and scanning electron microscope (SEM) technologies. The detailed preparation approaches for OM, EBSD and SEM observation can be found in Refs. [30,31]. Additionally, the detailed measuring method of the weight fraction of the δ phase can be seen in refs. [21,32].

## 3. Results and Discussion

### 3.1. Effects of Deformation Parameters on the Microstructure Evolution during Hot Deformation

#### 3.1.1. Effect of Deformation Temperature

From Figure 2b, it can be observed that there are plenty of long needle-like δ phases and intragranular δ phase nuclei after the strain-free aging treatment. Generally, defects in the polycrystal (such as dislocations, vacancies and gain boundaries) can serve as beneficial sites for δ phase nucleation [33,34]. In addition, there are few defects within the grain during the strain-free AT. Therefore, the nucleation of the δ phases on gain boundaries will precede that within grains [35,36]. Meanwhile, the precipitation of the δ phases at grain boundaries consumes plenty of Nb atoms, which further restrains the nearby intragranular nucleation of the δ phase and causes slow precipitation or growth of the intragranular δ phases [33].

Deformed microstructures with different deformation temperatures are shown in Figure 4. In Figure 4g–i, it can be seen that the weight fraction of the δ phase decreases with the increased deformation temperature. The weight fraction of the δ phase in the SEM image was gauged by the Image-Pro Plus 6 software (Media Cybernetics, Rockville, MD, USA). The weight fraction of the δ phase was measured five times using five different SEM figures. Eventually, the average number calculated from five SEM images is approximately represented the weight fraction of the δ phase of a working condition. The results show that the weight fraction of the δ phase decreased from 17.62% to 13.48% and then to 12.53% as the deformation temperature increased from 950 °C to 980 °C and 1010 °C. The reason is that the dissolution rate of the δ phase is closely associated with the temperature [37]. When the temperature rose above 980 °C, the δ phase began to dissolve in large quantities [38]. The remarkable thing is that there were some δ phase nuclei within deformed grains for the specimen deformed at 950 °C. However, there were almost no δ phase nuclei within the deformed grains for specimens deformed at 980 °C and 1010 °C. The reason is that high deformation temperatures can promote the quick dissolution of δ phase nuclei.

As demonstrated in Figure 4a–c, the recrystallization fractions at 950 °C and 980 °C were 22.31% and 18.01%, respectively, which are very close. However, when the deformation temperature increased to 1010 °C, the recrystallization fraction increased obviously, and its value was 32.10%. Additionally, it can be noticed that DRX grains form massively and grow significantly under high deformation temperatures. This shows that the high deformation temperature can accelerate the dissolution of the δ phase. Hence, the pinning effect of the δ phase on the growth of grains is weakened. In addition, high temperatures can also significantly induce recrystallization nucleation [39,40]. In general, the large kernel average misorientation (KAM) value means a high dislocation density [41]. As displayed in Figure 4d–f, the average KAM value reduced with the rise in deformation temperature. This is because the recrystallization and the dissolution of the δ phase consume deformation energy.

#### 3.1.2. Effect of Strain Rate

Figure 5 represents deformed microstructures with varying strain rates. According to Figure 4g and Figure 5e,f, the weight fraction of the δ phase declined from 17.62% to 16.87% and 10.98% as the strain rate decreased from 0.1 s^−1^ to 0.01 s^−1^ and 0.001 s^−1^. In addition, for the microstructure deformed at the strain rate of 0.001 s^−^^1^, the content of the unbroken long needle-like δ phases was lower than that at the high strain rate, and the content of the spherical and rod-like δ phases was higher. This phenomenon is mainly due to the high strain rate prolonging the dissolution time of the δ phases. Meanwhile, in Figure 4g and Figure 5e,f, the existence of intragranular δ phase nuclei in diverse strain rates can be seen.

Furthermore, as shown in Figure 4a and Figure 5a,b, it can be seen that the recrystallized fraction firstly increased from 22.31% to 31.85% and decreased to 25.91% as the strain rate declined from 0.1 s^−1^ to 0.01 s^−1^ and 0.001 s^−1^. In addition, the size of the DRX grains increased significantly with the decrease in the strain rate due to the prolongation of the recrystallized grain growth time at a low strain rate. Meanwhile, the average KAM value decreased with the reduction in strain rate (Figure 4d and Figure 5c,d).

#### 3.1.3. Effect of True Strain

Deformed microstructures with different true strains are depicted in Figure 6. As illustrated in Figure 4g and Figure 6e,f, the weight fraction of the δ phase reduced from 22.37% to 17.62% and 11.84% as the true strain increased from 0.36 to 0.69 and 1.20. The main cause of this is that the large strain accelerates the breakage of the δ phase and prolongs the dissolution time of the δ phase. For the specimen with the true strain of 1.20, it can be seen that the long needle-like δ phases are obviously bent, twisted and folded. This is because the large strain applied from the outside will result in the occurrence of a high-density dislocation area near the long needle-like δ phases, causing the deformation and fracture of the δ phases [42]. Meanwhile, a large quantity of dislocation around the long needle-like δ phases provides a fast path for the diffusion of Nb atoms, thereby accelerating the dissolution of the long needle-like δ phases during deformation [43,44]. Additionally, it is noteworthy that there are δ phase nuclei within deformed grains for the specimens with various true strains (Figure 4g and Figure 6e,f). This is because the δ phase nuclei formed during AT are not completely dissolved during hot deformation. However, the δ phase nuclei are extremely dense inside deformed grains for the sample deformed at the true strain of 1.20. The reason for this is that the large plastic deformation introduces a considerable quantity of crystal defects and energy storage, contributing to the dense nucleation of intragranular δ phases [45,46].

As depicted in Figure 4a and Figure 6a,b, the recrystallized fraction increased from 8.62% to 22.31% and 26.74% as the true strain raised from 0.36 to 0.69 and 1.20. In addition, the amount of DRX nuclei obviously increased as true strain increased, due to the promotion effect of the large strain on the continuous proliferation and accumulation of dislocations [47]. In addition, the average KAM value dropped with the rise of true strain (Figure 4d and Figure 6c,d).

### 3.2. Effects of Deformation Parameters on Microstructure Evolution during the First Annealing Stage

#### 3.2.1. Effect of Deformation Temperature

Figure 7 presents microstructures after the first annealing stage (i.e., AT) with different deformation temperatures. Compared to the deformed samples, the δ phase content of samples after the first annealing stage obviously increased. This is because lots of dislocations remained after deformation, thereby promoting the precipitation of the δ phase during AT [48]. From Figure 7g–i, it can be seen that the weight fraction of the δ phase first decreased and then increased. When the deformation temperature was 950 °C, 980 °C and 1010 °C, the weight fraction of the δ phase was 25.41%, 20.73% and 27.49%, respectively. According to ref. [28], deformation energy and DRX degree have a great influence on the precipitation of the δ phase, and the large DRX grain and high deformation energy can enhance the precipitation of the δ phase. Whereas the DRX degrees at 950 °C and 980 °C were close (Figure 4a,b), the specimen with the deformation temperature of 950 °C had high residual deformation energy (Figure 4d). Thus, the precipitation of the δ phase during AT is enhanced for the specimen with the deformation temperature of 950 °C. When the deformation temperature reached 1010 °C, the DRX degree increased significantly (Figure 4c). Although the deformation energy became lower (Figure 4f), the generation of lots of DRX grains increased the grain boundary area, thereby accelerating the precipitation of the δ phase.

Additionally, it can be seen that deformation temperature imposes significant influences on the morphology of the δ phase. For the sample with a deformation temperature of 950 °C, the δ phases formed during AT principally nucleate inside deformed grains and appear mainly in the form of clusters with rodlike morphology (Figure 7g). The reason for this is that there are some δ phase nuclei inside deformed grains after deformation (Figure 4g). Generally, the nucleation of the intragranular δ phase requires a long incubation time [35]. Therefore, the Nb atoms in the matrix during AT are preferentially used for the growth of the previous intragranular δ phase nuclei. In addition, the growth of the intragranular δ phase nucleus induces lattice distortion in the neighboring area, which can accelerate the adjacent δ phase nucleation afterward [35]. Generally, the content of Nb atoms in the matrix is fixed for a specific grade of superalloy. In addition, the Nb atom is an important constituent element of the δ phase (Ni_3_ Nb) [22]. Therefore, the content of the δ phase after deformation will affect the content of Nb atoms in the matrix, thereby influencing the size of the δ phase formed during AT. Because of the high content of the δ phase after deformation (Figure 4g), the Nb content in the matrix was low, which caused the small size of the δ phase formed during AT. Hence, the δ phases mainly occurred in the form of clusters with rodlike morphology in deformed grains after AT. However, as the deformation temperature was raised to 980 °C and 1010 °C, the size of the δ phase formed in AT became significantly larger, and the distribution became scattered (Figure 7h,i). This is because the δ phase nuclei are almost non-existent under high deformation temperature. Owing to the low content of the δ phase after deformation (Figure 4h,i), the Nb content in the matrix before AT was high, which increased the size of the the δ phase formed during AT. Moreover, compared to the sample deformed at 950 °C, the degree of DRX became larger and the distribution of Nb atoms became more uniform at high deformation temperature (Figure 4b,c), which promoted the precipitation and scattered distribution of the δ phase during AT.

As shown in Figure 4 and Figure 7, it can be seen that the recrystallized grains of samples deformed at different deformation temperatures after AT grew slightly compared with that before AT. The reason for this is that a large number of δ phases were precipitated during AT, thereby strongly restricting the growth of recrystallization grains. Additionally, as illustrated in Figure 7a–c, it can be seen that the recrystallization fractions at 950 °C and 980 °C were 35.54% and 34.47%, respectively, which are very close. However, when the deformation temperature was raised to 1010 °C, the recrystallization fraction increased to 58.57%. In addition, it is worth noting that when the deformation temperature reached 1010 °C, the recrystallized grain size after AT became significantly larger. This is because the content of the δ phase at the onset of AT was relatively low when the deformation temperature was 1010 °C (Figure 4i). Hence, the pinning effect of the δ phase was weakened in the early stage of AT, resulting in the further growth of previous large DRX grains. Furthermore, as the deformation temperature was elevated from 950 °C to 980 °C and 1010 °C, the average KAM value declined from 1.49° to 1.42° and 0.95°, respectively.

#### 3.2.2. Effect of Strain Rate

Figure 8 displays microstructures after AT with different strain rates. As depicted in Figure 7g and Figure 8e,f, the weight fraction of the δ phase first declined from 25.41% to 21.36% and then slightly increased to 22.52% as the strain rate dropped from 0.1 s^−1^ to 0.01 s^−1^ and 0.001 s^−1^. Apparently, the content of the δ phase of specimens deformed at the strain rates with 0.01 s^−1^ and 0.001 s^−1^ after AT is almost the same. However, the morphology of the δ phase was significantly diverse at different strain rates. For the specimens with the strain rates of 0.1 s^−1^ and 0.01 s^−1^, the rodlike δ phase precipitated during AT occurred mainly in the form of clusters inside the deformed grains. This is because the specimens deformed at strain rates of 0.1 s^−1^ and 0.01 s^−1^ have a higher residual content of δ phases after deformation (Figure 4g and Figure 5f), resulting in a lower content of Nb element in the matrix, and therefore the size of the δ phase formed during AT is small. However, the existence of intragranular δ phase nuclei after deformation promoted the precipitation of the δ phase clusters within deformed grains during AT. Nevertheless, when the strain rate was 0.001 s^−1^, plenty of δ phases were densely generated inside deformed grains with elongated rodlike morphology during AT (Figure 8e). The major cause for this is that the content of the δ phase after deformation was extremely low (Figure 5e), which means that the content of Nb atoms incorporated into the matrix increased, causing the occurrence of elongated rodlike δ phases formed during AT. In addition, there were still some δ phase nuclei after deformation (Figure 5f), which promoted the precipitation of dense intragranular δ phases.

As presented in Figure 7a and Figure 8a,b, the recrystallization fraction after AT first increased from 35.54% to 39.05% and then decreased to 23.52% as the strain rate declined from 0.1 s^−1^ to 0.01 s^−1^ and 0.001 s^−1^. At the same time, the average KAM value was 1.49°, 1.26° and 1.37°, respectively (Figure 7d and Figure 8c,d).

#### 3.2.3. Effect of True Strain

Figure 9 illustrates microstructures after AT with different true strains. As demonstrated in Figure 7g and Figure 9e,f, the weight fraction of the δ phase descended from 31.72% to 25.41% and 14.44% when the true strain was elevated from 0.36 to 0.69 and 1.20. When the true strain was 0.36, the δ phases formed during AT precipitated in the form of clusters with spherical morphology within the deformed grains, which is similar to the precipitation characteristics of the specimen with a true strain of 0.69 (Figure 7g and Figure 9e). The reason for this is that the content of the δ phase after deformation was extremely high (Figure 6e), which means that the content of Nb atoms in the matrix was extremely small. At the same time, there were many intragranular δ phase nuclei after deformation, and the low Nb atom content preferentially promotes the growth of these phase nuclei, resulting in the appearance of δ phase clusters with spherical morphology. However, the morphology and distribution of the δ phases varied obviously when the true strain increased to 1.20. There were large numbers of spherical and rodlike δ phases in deformed grains. This is because of the low content of the δ phase after deformation (Figure 6f) caused an increase in the number of Nb atoms incorporated into the matrix. However, these Nb atoms were simultaneously used for the growth of an excess number of intragranular the δ phase nuclei generated during deformation, resulting in the limited growth of the δ phases during AT. Hence, the δ phase after AT was densely precipitated within deformed grains in the form of spherical and rod-like morphology.

As depicted in Figure 7a and Figure 9a,b, the recrystallization fraction after AT increased from 23.93% to 35.54% and 61.37% as the true strain increased from 0.36 to 0.69 and 1.20. In addition, the average KAM value was same (1.49°) for the specimen with the true strain of 0.36 and 0.69. However, the average KAM value declined to 1.05° when the true strain increased to 1.20.

### 3.3. Effects of Deformation Parameters on Microstructure Evolution during the Second Annealing Stage

To further compare the microstructure after CRT with diverse deformation parameters, the average grain size ($\bar{d}$) and heterogeneous factor (f_h_) are used. As reported by Chen et al. [49], the heterogeneous factor can be introduced to quantitatively describe the uniformity of microstructure, as showed by the equation below:(1)$$f_{h}=\frac{\sum_{i=1}^{i=N}|d_{i}-\bar{d}|}{N\bar{d}}$$
where N is the amount of counted grains, d_i_ is the size of the grain i, and $\bar{d}$ is the average grain size. Based on the definition, the lower f_h_ value represents the more homogeneous microstructure.

#### 3.3.1. Effect of Deformation Temperature

Figure 10 represents microstructures after the second annealing stage (i.e., CRT) with varying deformation temperatures. As presented in Figure 10a–c, it can be found that the recrystallized degree after CRT firstly increased and then declined with the increase of deformation temperature. In addition, $\bar{d}$ and f_h_ firstly decreased and then increased with the elevation of deformation temperature. Especially for the sample whose deformation temperature is 980 °C, the recrystallization after CRT was almost complete, and $\bar{d}$ and f_h_ are 6.53 μm and 0.48, respectively. However, there were still many deformed grains in the samples with deformation temperatures of 950 °C and 1010 °C. The main cause for this is that the content, distribution and morphology of the δ phases and deformation energy before CRT are different under diverse deformation temperatures.

For the sample deformed at 950 °C, deformed grains contain clusters of δ phases with a rodlike shape after AT (Figure 7g). Owing to the short phase interval, the dislocation motion was hindered during CRT, which is not beneficial to the continuous accumulation of dislocation [50]. Hence, the recrystallization nucleation was slow. Meanwhile, the high content of the δ phase limited the growth of recrystallized grains, thereby causing the existence of many deformed grains after CRT. It is worth noting that a large deformed grain was divided into two deformed grains by fine recrystallized grains arranged in a linear pattern, as presented in Figure 10d. In addition, the deformation energy near the recrystallized grains arranged in a linear pattern was high, but the deformation energy inside the deformed grain was low. The reason is that there was a long needle-like δ phase running through the deformed grain and clusters of intragranular δ phases with rodlike morphology after AT (see Figure 7g). In the early stage of CRT, although the high temperature is conducive to the movement of dislocation, the extremely dense intragranular the δ phases hindered the movement of dislocations. As the annealing progresses, these dense intragranular δ phases gradually dissolve, while the long needle-like δ phases dissolve slowly. Under the stimulation of thermal energy, the dislocations began to move and were blocked by the long needle-like δ phase and the grain boundary. Hence, the energy inside the deformed grain was low, and the energy in the vicinity of the long needle-like δ phase and grain boundary was high. The high deformation energy around the long needle-like δ phase promoted the SRX nucleation, thereby dividing the large deformed grain into two deformed grains. In addition, it can be also seen that some recrystallized grains were formed inside the deformed grain (Figure 10d). This is because the rod-like δ phases resulting from the dissolution of the long needle-like δ phase during CRT enhanced the intragranular SRX nucleation.

Since the deformation energy was nearly the same before CRT (Figure 7d,e), the change of recrystallized degree between samples deformed at 950 °C and 980 °C after CRT was mainly affected by the content, distribution and morphology of the δ phases before CRT. For the samples deformed at 980 °C, the δ phase after AT was dominated by scattered δ phases with rodlike and needle-like shapes (Figure 7h). In general, the large second-phase particles can impede the movement of dislocation, thereby promoting the formation of high-density dislocation cells. These dislocation cells accelerate SRX nucleation, which is also called particle stimulation nucleation (PSN) [51,52]. Therefore, the dispersed large δ phase was conducive to recrystallization nucleation during CRT. In addition, the initial δ phase content at the beginning of the CRT was 20.73%, lower than that of 950 °C and 1010 °C, causing a weak pinning effect on the growth of grains. Thus, the overall recrystallization was close to completion. In addition, the average KAM value after the CRT was low, at only 0.52° (Figure 10e).

For the samples deformed at 1010 °C, although the δ phase after AT was also dominated by scattered δ phases (Figure 7i), the KAM value at the beginning of the CRT was low (Figure 7f), at only 0.95°, thereby slowing down the SRX nucleation. In addition, the δ phase content was as high as 27.49% before CRT, which greatly inhibited the growth of the recrystallized grains. Consequently, some deformed grains remained in the microstructure. The remarkable thing is that there are more recrystallized grains formed inside the deformed grain compared to that of the deformation of 950 °C, as presented in Figure 10d,f. The main cause of this is that the evenly distributed δ phase is mainly needle-like and rod-like in the sample with the deformation temperature of 1010 °C, thereby providing more positions of SRX nucleation in deformed grains.

In Figure 10g–i, it can be seen that the content of the δ phase significantly declined after CRT. As the deformation temperature increased from 950 °C to 980 °C and 1010 °C, the residual δ phase content after CRT was 4.43%, 3.55% and 6.26%, respectively. For the samples with the deformation temperatures of 950 °C and 1010 °C, the morphology of the δ phase were mainly needle-like and rod-like. Nevertheless, as for the sample deformed at 980 °C, the δ phase was primarily rod-like and spherical.

#### 3.3.2. Effect of Strain Rate

Figure 11 demonstrates the microstructures after the second annealing stage with different strain rates. As illustrated in Figure 10a and Figure 11a,b, $\bar{d}$ and f_h_ after CRT first declined and then slightly raised as the strain rate decreased. Additionally, it can be seen that the microstructure with the strain rate of 0.01 s^−1^ contained fewer deformed grins compared to that of 0.1 s^−1^ and 0.001 s^−1^, which means that the recrystallized degree was higher in the annealed microstructure with the strain rate of 0.01 s^−1^.

For the sample deformed at the strain rate of 0.01 s^−1^, the degree of recrystallization after AT was high compared to the samples with the strain rates of 0.1 s^−1^ and 0.001 s^−1^ (Figure 7a and Figure 8a,b). In addition, the δ phases after AT primarily occurred inside deformed grains in the form of clusters (Figure 8f). That means that the SRX nucleation in the early stage of CRT was difficult, due to the short interval between the intragranular δ phases [50]. However, the size of the intragranular δ phases was relatively small, causing the quick dissolution of the intragranular δ phases during CRT. This makes the growth of previously formed recrystallized grains difficult to pin. Meanwhile, owing to the lack of growth time, some small recrystallized grains which were newly nucleated during CRT were retained. Hence, the recrystallized degree after CRT was high. The remarkable thing is that there was a large deformed grain after CRT, as presented in Figure 11d. The reason for this is that the δ phases precipitated in the form of clusters during AT were not completely dissolved during CRT. As depicted in Figure 11f, many shortened intragranular the δ phases were retained in some deformed grains. That is not conducive to the occurrence of SRX nucleation inside deformed grains. Therefore, this deformed grain can only be slowly swallowed by the recrystallized grains outside the deformed grain.

For the sample deformed at the strain rate of 0.001 s^−1^, S_15–30_ reaches 0.21 (Figure 11a), which is higher than that of the sample with the strain rate of 0.01 s^−1^ (Figure 11b). This is because the size of plentiful recrystallized grains before CRT for the sample with the strain rate of 0.001 s^−1^ was large compared to that of 0.01 s^−1^. Additionally, the residual deformation energy after CRT was high, and the average KAM value reached 1.01°, as illustrated in Figure 11c. In addition, it can be seen that some large deformed grains were split into small deformed grains by recrystallized grains. This is because the microstructure before CRT contained many long needle-like δ phases and dense intragranular δ phases with elongated rodlike morphology, which are favorable to the SRX nucleation during CRT. For example, some large recrystallized grains are formed inside deformed grains. However, those δ phases dissolve slowly during CRT (Figure 11e), thereby limiting the growth of the recrystallized grains. Hence, many deformed grains were retained after CRT.

As displayed in Figure 10g and Figure 11e,f, the residual δ phase content after CRT was, respectively, 4.43%, 4.62% and 6.07%, as the strain rate declined from 0.1 s^−1^ to 0.01 s^−1^ and 0.001 s^−1^. As for the samples with strain rates of 0.1 s^−1^ and 0.001 s^−1^, the δ phase was mainly needle-like and rod-like. Meanwhile, the morphology of the the δ phases were primarily rodlike and spherical for the sample deformed at the strain rate of 0.01 s^−1^.

#### 3.3.3. Effect of True Strain

Figure 12 depicts the microstructure after the second annealing stage with different true strains. The $\bar{d}$ and fh after CRT first increased and then obviously declined with the rise of true strain, as displayed in Figure 10a and Figure 12a,b. Furthermore, as shown in Figure 10d and Figure 12c,d, it can be seen that some large deformed grains still existed in the microstructures with true strains of 0.36 and 0.69. However, the recrystallization of the sample with the true strain of 1.20 was nearly complete, and the average KAM value was as low as 0.59°.

For the specimen deformed at the true strain of 0.36, the degree of recrystallization after AT was lower than that of the sample with the true strain of 1.20 (Figure 9a,b). Additionally, there was a large quantity of long needle-like δ phases and clusters of intragranular δ phases with spherical morphology before CRT, as presented in Figure 9e. The distribution and morphology of the δ phase inhibited the intragranular SRX nucleation in the early stage of CRT and the growth of previously formed recrystallized grains during CRT, thereby slowing down the recrystallization. Hence, there were still many deformed grains after CRT. In addition, there were dense shortened intragranular δ phases after CRT, as depicted in Figure 12e. That proves from the side why there are large deformed grains after CRT. As demonstrated in Figure 12c, it is noteworthy that recrystallized grains arranged in the linear pattern appeared inside the deformed grains. The main cause for this is that there were many long needle-like the δ phases running through deformed grain before CRT, causing the SRX nucleation along long needle-like phases boundary during CRT.

As for the specimen with the true strain of 1.20, there was an excess number of recrystallized grains after AT (Figure 9b). Moreover, the content of the δ phase was as low as 14.44%, and there were many small spherical and rodlike the δ phases before CRT (Figure 9f). In general, the dispersed small second-phase particles have a strong pinning effect on the grain boundaries, hindering the growth of grains [53,54]. Hence, the growth rate of previously formed massive recrystallized grains was restricted during CRT, thereby obtaining an even microstructure with an average grain size of 5.93 μm. Furthermore, the spherical δ phase dominated the retained δ phase after CRT, as presented in Figure 12f. That again proves that the growth of recrystallized grains was limited during CRT.

As displayed in Figure 10g and Figure 12e,f, as the true strain increased from 0.36 to 0.69 and 1.20, the residual δ phase content after CRT was 6.07%, 4.43% and 4.35%, respectively. For the samples with the true strain of 0.36 and 0.69, the δ phase were mainly needle-like and rod-like. However, the spherical δ phase dominated for the sample deformed at the true strain of 1.20.

## 4. Conclusions

In this investigation, firstly, the hot compressive tests with different deformation parameters were completed. Subsequently, the effect of deformation parameters on microstructure evolution during a new two-stage annealing process was analyzed. The main conclusions are as follows:

(1) The precipitation and dissolution of small size intragranular-δ phase (i.e., intragranular-δ phase nuclei) after deformation have a great influence on the distribution of the δ phase during AT. When the dissolution of the intragranular-δ phase nuclei is complete under a high deformation temperature, the dispersion distribution of δ phase forms during AT. However, when the dissolution of the intragranular-δ phase nuclei is relatively weak compared to the precipitation behavior during deformation, the clusters of the intragranular δ phases will generate during AT.

(2) The morphology of the δ phases forming during AT is strongly related to the retained content of the δ phases after deformation. The size of the δ phases will be large and the length of the δ phases will be long during AT when the retained content of the δ phases is low after deformation. The reason is that there is an equilibrium of the content of the δ phase and Nb atom. The Nb atom content in the matrix becomes high as the content of the δ phase decreases during deformation, resulting in more Nb atoms retained for the precipitation of the the δ phase during AT.

(3) The distribution and morphology of the δ phase after AT have a more positive impact on the elimination of the mixed grain microstructure during CRT compared to the deformation energy. The reason is that the distribution and morphology of the δ phase play a leading role in the thermal motion of dislocation and grain boundary. The clusters of intragranular δ phases with spherical and rodlike shapes after AT hinder the dislocation movement within deformed grains during CRT, which is bad for the accumulation of dislocation. Thus, the recrystallization nucleation rate within deformed grains is low, thereby slowing down the elimination of the deformed grains. However, the dispersion distribution of the δ phase with rodlike and needle-like morphology after AT is good for the formation of high-density dislocation cells during CRT. Consequently, the annealed microstructure is uniformly refined due to the strong PSN effect of those large δ phases.

(4) The refining mechanism for the deformed samples of diverse deformation parameters during annealing is different and can be summarized as follows: High deformation temperature can promote the scattered distribution of the δ phase during AT, thereby evenly refining the mixed grain microstructure to an average grain size of 6.52 μm after CRT. A large true strain can contribute to the occurrence of a large quantity of DRX nuclei and the precipitation of many small the δ phases. Hence, the growth of those DRX grains is efficiently pinned during CRT. Eventually, an even microstructure with an average grain size of 5.93 μm is obtained. Additionally, a high recrystallization degree and proper deformation energy can be retained in the microstructure with a suitable strain rate, which helps to refine the mixed grain microstructure to an average grain size of 8.63 μm after CRT.

## Figures and Tables

**Figure 1 materials-15-05508-f001:**
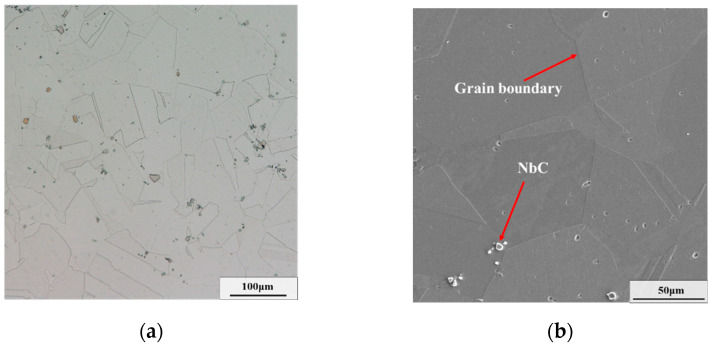
Microstructure of GH4169 superalloy after the solution treatment: (**a**) OM map; (**b**) SEM map.

**Figure 2 materials-15-05508-f002:**
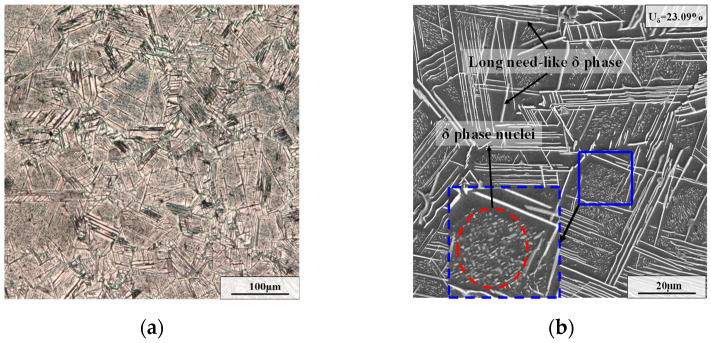
Microstructure of GH4169 superalloy after aging treatment: (**a**) OM map; (**b**) SEM map. (U_δ_ represents the weight fraction of the δ phase).

**Figure 3 materials-15-05508-f003:**
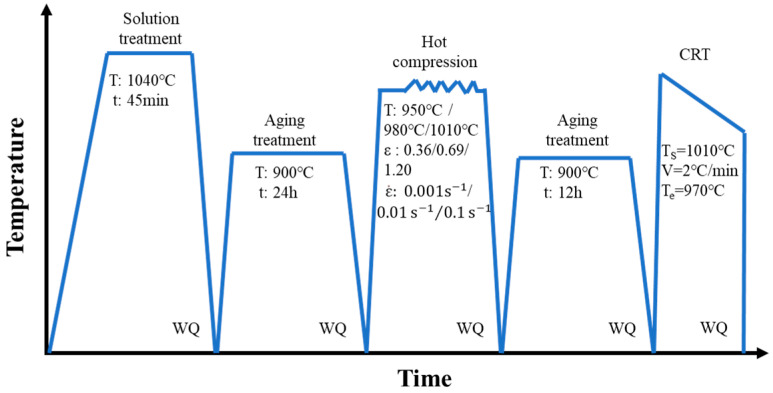
Experimental process for the whole test (WQ represents water quenching).

**Figure 4 materials-15-05508-f004:**
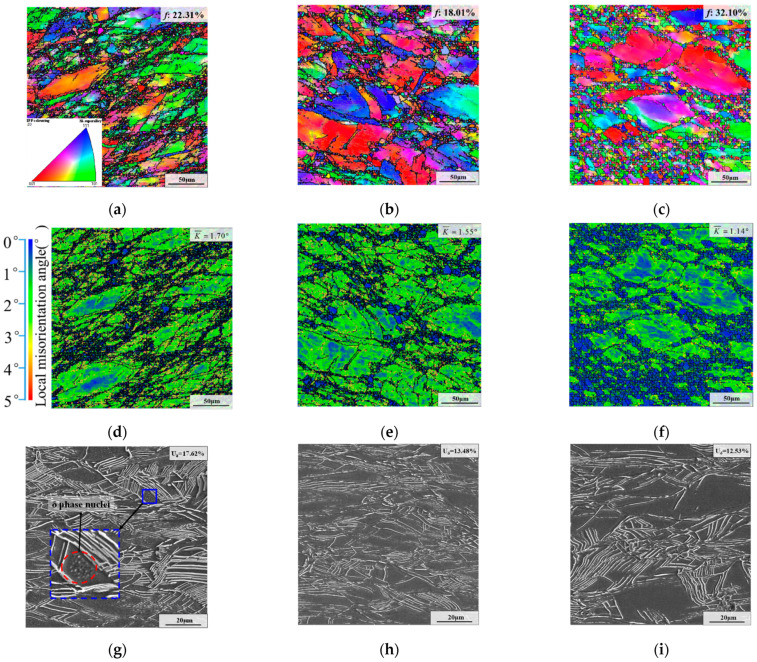
Deformed microstructures with different deformation temperatures of: (**a**,**d**,**g**) 950 °C; (**b**,**e**,**h**) 980 °C; (**c**,**f**,**i**) 1010 °C. (The strain rate is 0.1 s^−1^, and the true strain is 0.69. $\bar{K}$ represents the average KAM value, and *f* represents the recrystallized fraction).

**Figure 5 materials-15-05508-f005:**
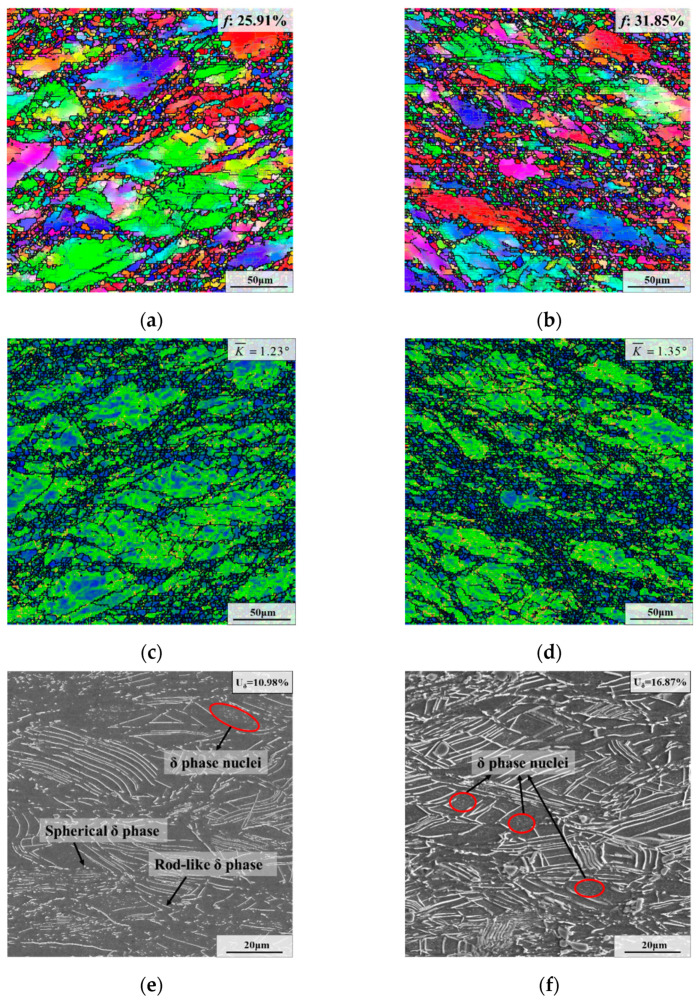
Deformed microstructures with varying strain rates of: (**a**,**c**,**e**) 0.001 s^−1^; (**b**,**d**,**f**) 0.01 s^−1^. (The deformation temperature is 950 °C, and the true strain is 0.69.)

**Figure 6 materials-15-05508-f006:**
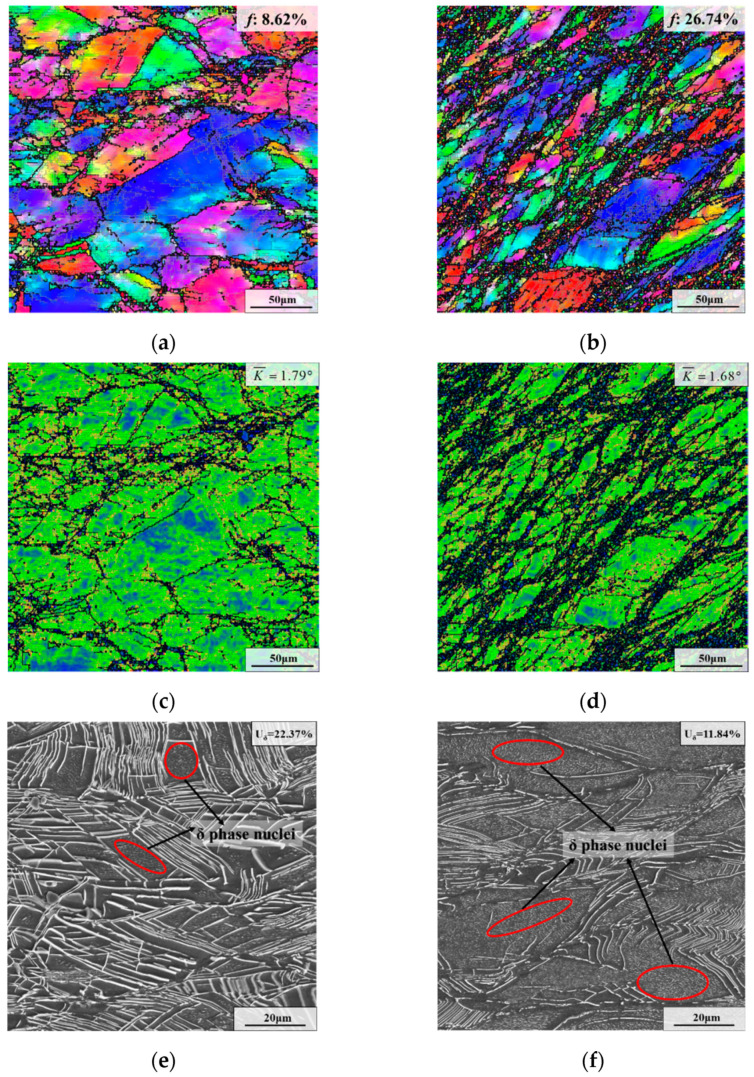
Deformed microstructures with different true strains: (**a**,**c**,**e**) 0.36; (**b**,**d**,**f**) 1.20. (The deformation temperature is 950 °C, and the strain rate is 0.1 s^−1^).

**Figure 7 materials-15-05508-f007:**
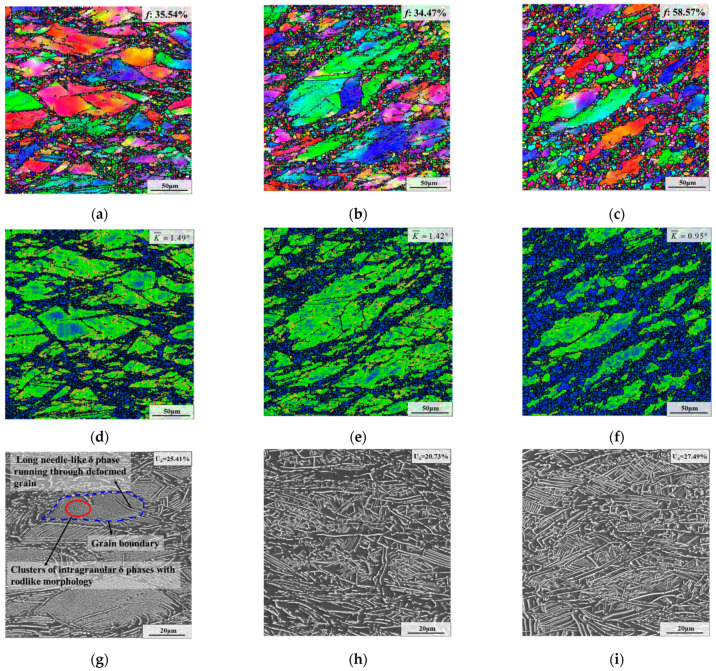
Microstructures after the first annealing stage with different deformation temperatures of: (**a**,**d**,**g**) 950 °C; (**b**,**e**,**h**) 980 °C; (**c**,**f**,**i**) 1010 °C. (The strain rate is 0.1 s^−1^, and the true strain is 0.69).

**Figure 8 materials-15-05508-f008:**
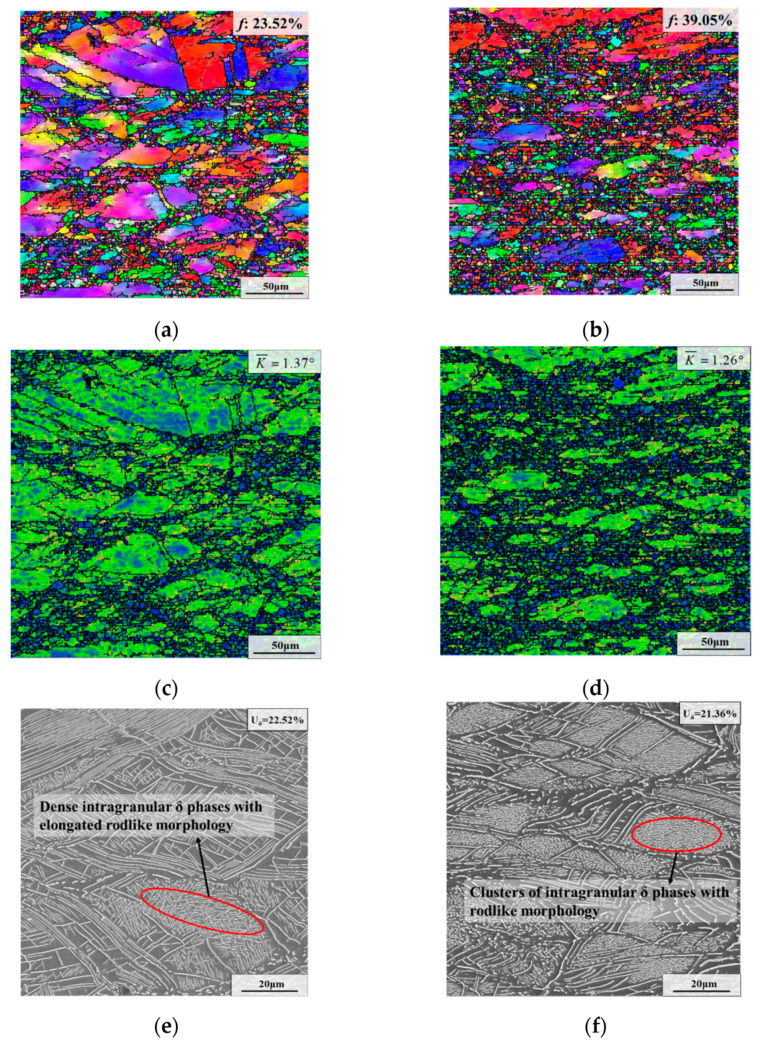
Microstructures after the first annealing stage with varying strain rates of: (**a**,**c**,**e**) 0.001 s^−1^; (**b**,**d**,**f**) 0.01 s^−1^. (The deformation temperature is 950 °C, and the true strain is 0.69).

**Figure 9 materials-15-05508-f009:**
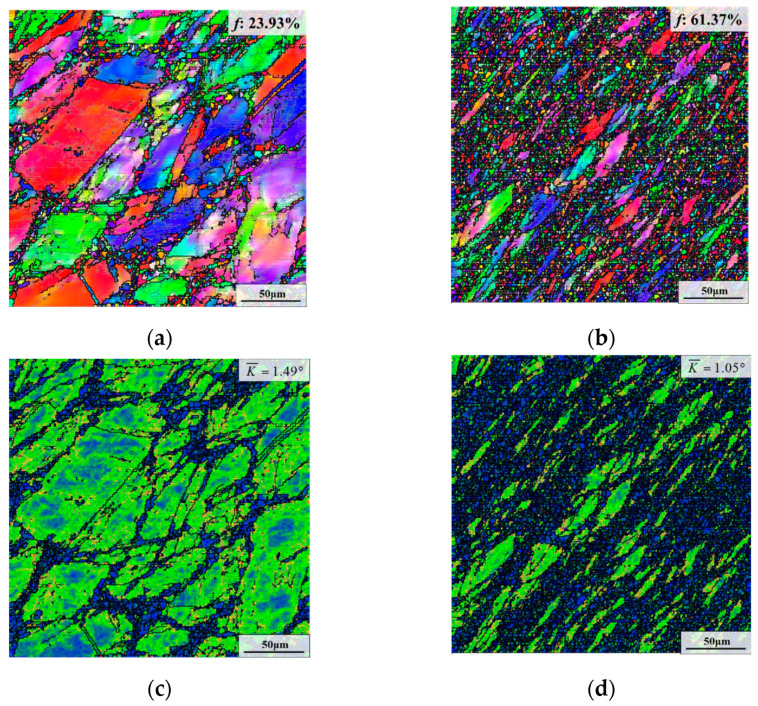
Deformed microstructures after the first annealing stage with different true strains: (**a**,**c**,**e**) 0.36; (**b**,**d**,**f**) 1.20. (The deformation temperature is 950 °C, and the strain rate is 0.1 s^−1^).

**Figure 10 materials-15-05508-f010:**
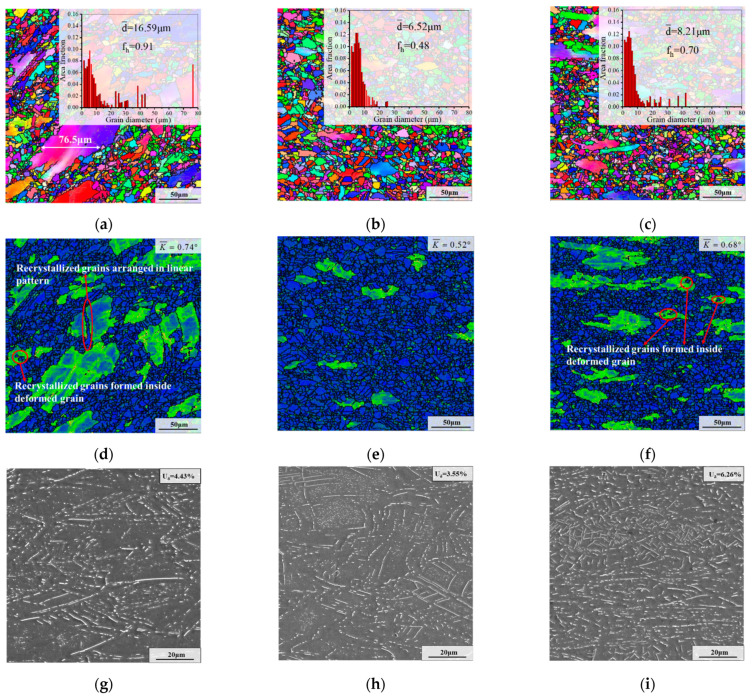
Microstructures after the second annealing stage with different deformation temperatures of: (**a**,**d**,**g**) 950 °C; (**b**,**e**,**h**) 980 °C; (**c**,**f**,**i**) 1010 °C. (The strain rate is 0.1 s^−1^, and the true strain is 0.69).

**Figure 11 materials-15-05508-f011:**
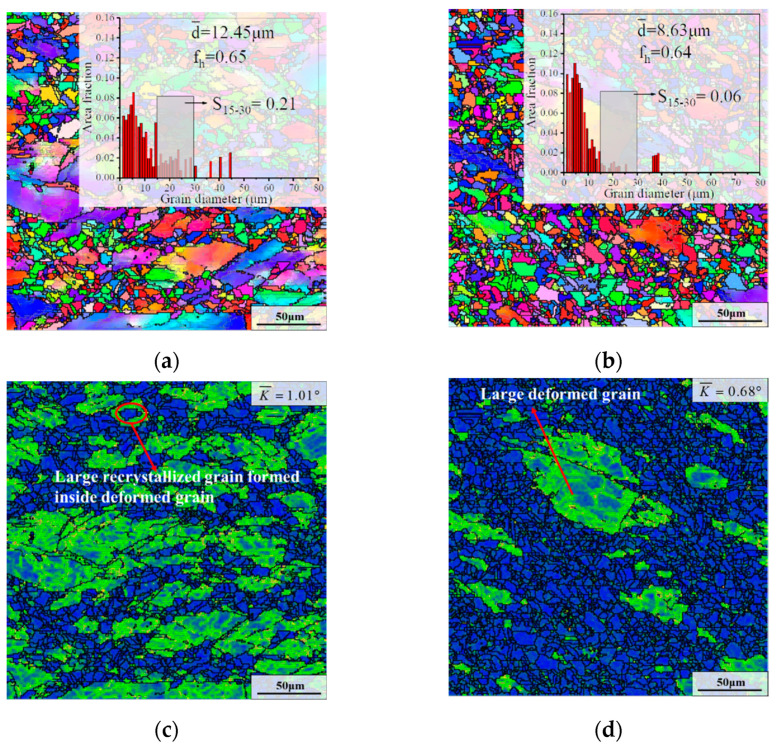
Microstructures after the second annealing stage with various strain rates of: (**a**,**c**,**e**) 0.001 s^−1^; (**b**,**d**,**f**) 0.01 s^−1^. (The deformation temperature was 950 °C, and the true strain was 0.69. S_15–30_ represents the cumulative value of the area fraction of average grain size in the range of 15–30 μm).

**Figure 12 materials-15-05508-f012:**
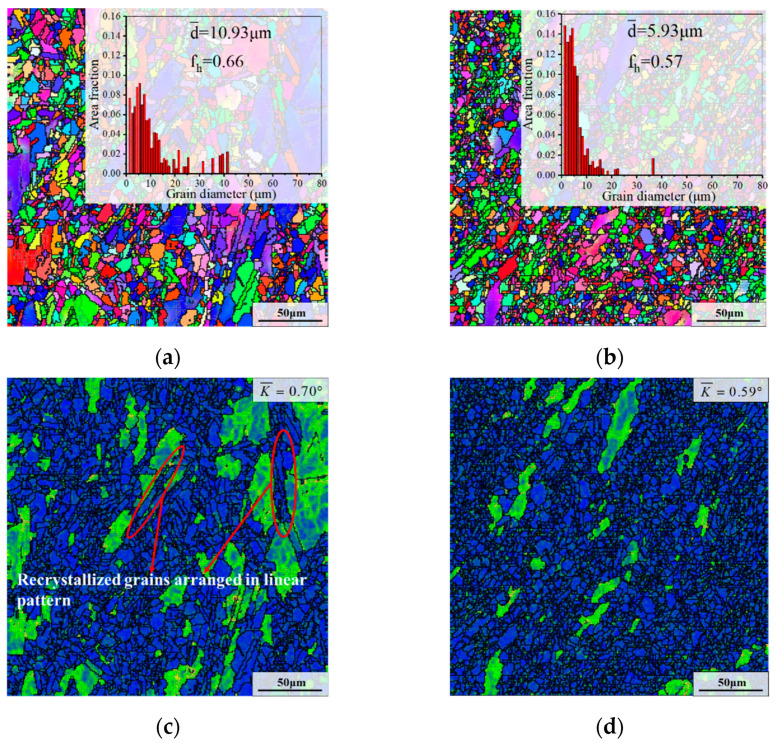
Microstructures after the second annealing stage with different true strains: (**a**,**c**,**e**) 0.36; (**b**,**d**,**f**) 1.20. (The deformation temperature was 950 °C; and the strain rate was 0.1 s^−1^).

**Table 1 materials-15-05508-t001:** Chemical composition of studied GH4169 superalloy.

Element	Ni	Cr	Nb	Mo	Ti	Al	Co	C	Fe
**Content (wt.%)**	52.82	18.96	5.23	3.01	1.00	0.59	0.01	0.01	bal.

**Table 2 materials-15-05508-t002:** Experimental scheme of hot deformation.

Scheme	Deformation Parameters		
	Temperature/°C	Strain Rate/s^−1^	True Strain
1	950	0.1	0.69
2	980	0.1	0.69
3	1010	0.1	0.69
4	950	0.001	0.69
5	950	0.01	0.69
6	950	0.1	0.36
7	950	0.1	1.20

## Data Availability

The raw/processed data required to reproduce these findings cannot be shared at this time as the data also form part of an ongoing study.

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
