# Peer review of "Effect of Deformation Parameters of an Initial Aged GH4169 Superalloy on Its Microstructural Evolution during a New Two-Stage Annealing"

_materials, 2022, doi:10.3390/ma15165508_

Round 1

Reviewer 1 Report

his work is the extension of the authors previous work. In their previous work on the same material [published in J. Alloys. Compd. 907 (2022)164] authors found that by using two-stage heat treatment composed of At & CRT, initial aged deformed GH4169 superalloy can be homogeneously refined.  In this paper they have deeply investigated the role of deformation parameters with the evolution of microstructures during the two-stage annealing. Authors found during cooling recrystallization annealing (CRT) treatment after aging treatment the deformation parameters largely affect the grain microstructure by modifying the content, distribution and morphology of Ni3Nb phase after deformation due to the equilibrium of the content of Ni3Nb phase and Nb atoms in the material. The experimental analysis, discussion and conclusions are very rigorous and of high interest to the community of modern energy and aerospace industries due to fine corrosion resistance and outstanding mechanical properties of GH4169 superalloy. To my understanding it is worth publishing in this journal.

Author Response

 Thanks for the reviewer’s constructive comments. Acorrding to the constructive comments, the manuscript has been carefully revised! Thanks!

Reviewer 2 Report

The submitted paper is devoted to the topic of effect of deformation parameters on a microstructure changes of the GH4169 superalloy. The quality of the paper is sufficient and the results are nicely presented. There are just few remarks I would like the Authors to answer:

1. Authors discuss the phases, however, there should be some proves of their chemical composition, e.g. results obtained by TEM, EDS or XRD analysis should be provided.

2. Authors provide EBSD maps. It be of interest to add also data of texture components and discuss if/how it changes together with microstructure.

3. From EBSD maps it can be seen that many points have not been indexed. Authors are asked to explain it and how it also changed for each state.

Reviewer 3 Report

The manuscript describes “Effect of deformation parameters of an initial aged GH4169 superalloy on its microstructural evolution during a new two-stage annealing”, which can be suitable for Materials. Anyhow, the reviewer would like to make the following comments;

Abstract

1. Rewrite the following sentences and remove the cited reference.

“In the authors’ previous work, it could be observed that the uneven deformed microstructure of an initial aged deformed GH4169 superalloy can be homogeneously refined through a new two-stage annealing way composed of an aging treatment (AT) and a cooling recrystallization annealing treatment (CRT) [J. Alloy. Compd., 907(2022)164-325].”

Introduction

1. Follow the journal template for your manuscript.

2. Introduction has to be improved and discussed in detail for cited references.

Materials and Experiments

1. Present the chemical composition and mechanical properties of the based material in one table.

2. How to justify different heat treatment processes;

·         “To dissolve all precipitates into the matrix and acquire homogenous initial microstructures, these samples firstly were solution-treated (T=1040 C, t =45min)”

·         “Then, an aging treatment (T=900 C, t=24h) was performed to get sufficient δ phases”

·         “the two-stage annealing treatment which consists of an AT for precipitating some δ phases and a subsequent CRT to refine deformed mixed grains was implemented”

·         Heat treatment processes in Figure 3.

3. Present the all process parameters in one table.

Results and discussion

1. “Deformed microstructures with different deformation temperatures are shown in Figure 4. From Figure 4g-i, it can be found that the weight fraction of the δ phase decreases from 17.62% to 13.48% and then to 12.53% as the deformation temperature increases from 950 C to 980 C and 1010 C .”

Figure 4(g-I) does not present the weight fraction reduction of the δ phase.

2. Mention the effect of process parameters on average grain size, recrystallized fraction, and fraction of δ phase with values in sections 3.1 and 3.2.

Round 2

Reviewer 3 Report

The authors have responded to the reviewer's concerns one by one. The revised version is accepted for publication.